# Exploring perceptions and operational considerations for use of a smartphone application to self-monitor blood pressure in pregnancy in Lombok, Indonesia: protocol for a qualitative study

Tigest Tamrat ,[1,2] Yuni Dwi Setiyawati,[3] Maria Barreix,[1] Mergy Gayatri,[3,4] Shannia Oktaviana Rinjani,[3] Melissa Paulina Pasaribu,[3] Antoine Geissbuhler,[5] Anuraj H Shankar,[3,6] Özge Tunçalp[1]

For numbered affiliations see end of article.

**Correspondence to**
Tigest Tamrat; tamratt@who.int

## ABSTRACT

**Introduction** Hypertensive disorders of pregnancy (HDP) are a leading cause of maternal deaths globally and require close monitoring of blood pressure (BP) to mitigate potential adverse effects. Despite the recognised need for research on self-monitoring of blood pressure (SMBP) among pregnant populations, there are very few studies focused on low and middle income contexts, which carry the greatest burden of HDPs. The study aims to understand the perceptions, barriers, and operational considerations for using a smartphone software application to perform SMBP by pregnant women in Lombok, Indonesia.

**Methods and analysis** This study includes a combination of focus group discussions, in-depth interviews and workshop observations. Pregnant women will also be provided with a research version of the smartphone BP application to use in their home and subsequently provide feedback on their experiences. The study will include pregnant women with current or past HDP, their partners and the healthcare workers involved in the provision of antenatal care services within the catchment area of six primary healthcare centres. Data obtained from the interviews and observations will undergo thematic analyses using a combination of both inductive and deductive approaches.

**Ethics and dissemination** The study was approved by the World Health Organization (WHO) and Human Reproduction Programme (HRP) Research Project Review Panel and WHO Ethical Review Committee (A65932) as well as the Health Research Ethics Committee, Faculty of Medicine, Universitas Mataram in Indonesia (004/UN18/F7/ETIK/2023).
Findings will be disseminated through research publications and communicated to the Lombok district health offices. The analyses from this study will also inform the design of a subsequent impact evaluation.

## STRENGTHS AND LIMITATIONS OF THIS STUDY

⇒ This study addresses a research gap identified by the WHO and expands on the geographical scope of available evidence on self-monitoring of blood pressure (SMBP) beyond high-income settings.
⇒ The study leverages a novel approach of a smartphone-based blood pressure (BP) measurement to examine the intersection of digital health, maternal health, self-care and health systems.
⇒ To provide a holistic perspective, the study will obtain insights from a diverse set of participants, including pregnant women, their partners and the health workforce across different levels of the health system (e.g. community health workers, midwives, doctors, managers).
⇒ A limitation of this study is that pregnant women will not be actively self-monitoring their BP to detect clinical changes but rather undergo the process to understand their perceptions of SMBP and implementation considerations.
⇒ This study will inform subsequent impact evaluations and complements existing initiatives within the research site on the use of digital tools towards identification and management of hypertensive disorders of pregnancy.

## INTRODUCTION

Hypertensive disorders of pregnancy (HDPs) are one of the most common pregnancy complications and a leading cause of maternal mortality and morbidity.[1–5] Approximately 14% of all maternal deaths globally are attributed to HDPs, with the majority in low and middle-income contexts.[2 3] HDPs include chronic hypertension, gestational hypertension, pre-eclampsia and chronic hypertension with superimposed pre-eclampsia.[1 5] Chronic hypertension consists of elevated blood pressure (BP) that is identified prior to the pregnancy or detected within the first 20 weeks.[1] Gestational hypertension is considered to be induced during a pregnancy with the onset of elevated BP after 20 weeks.[1] Pre-eclampsia

is characterised by elevated BP in the presence of either proteinuria or other new onset maternal organ dysfunction, neurological conditions or fetal growth restriction.[6] It is a precursor to eclampsia, a life-threatening condition which accounts for the majority of deaths attributed to HDPs.[2 4]

The World Health Organization (WHO) recommends routine BP measurement during pregnancy and at every point of contact in the provision of antenatal care (ANC) as part of efforts to improve maternal health.[7 8] In certain contexts, pregnant women identified with HDP may require additional follow-up and monitoring in order to mitigate potential adverse outcomes.[9] Furthermore, as the underlying cause of pre-eclampsia is unknown and onset may be triggered in between clinical consultations, close monitoring of BP beyond the health facility is critical.[5 10 11]

## Self-monitoring of blood pressure

Self-monitoring of blood pressure (SMBP), also known as home-based BP monitoring, is well researched in the general population[12–17] with a WHO recommendation in 2013 supporting its appropriate and affordable use.[18] Cited advantages include improved BP control and reduced anxiety associated with 'white coat' hypertension.[10–12 15 17] However, studies also highlight challenges associated with the variety of devices used within home settings, reliability of readings and calibration requirements, as well as ensuring guidance to individuals on interpreting results, dealing with variations of readings and appropriate actions to be taken.[19 20]

SMBP during pregnancy is an emerging approach recommended by WHO in 2021 'as an additional option to clinic blood pressure monitoring by healthcare workers during antenatal contacts only, for individuals with hypertensive disorders of pregnancy'.[21] Although there is a substantial literature base on SMBP in non-pregnant populations,[12–16 19 20 22–29] the evidence specific to pregnancy is still nascent[11 30–33] and limited for low- and middle-income countries (LMIC) where there remains a high burden of HDPs.[34] To date, studies that explore SMBP among pregnant women span Australia, Canada, Japan, UK, New Zealand and USA,[35 36] with one study in South Africa focused on knowledge and attitudes of SMBP among patients with pre-eclampsia.[34] A Cochrane systematic review on the different approaches to SMBP in pregnancy indicated uncertainties in the benefits of SMBP of hypertensive pregnancies due to limited number of studies.[37] Other systematic reviews on SMBP during antenatal and postpartum periods demonstrated its feasibility but no difference in clinical outcomes,[38] while another meta-analyses reported reductions in ANC visits, hospital admissions and diagnosis of pre-eclampsia.[39] All systematic reviews highlighted the lack of studies from LMICs and the need for more evidence, particularly across lower resource settings.[37–39]

Among the available studies, the Blood Pressure Self-Monitoring in Pregnancy (BuMP) research initiative conducted in the UK, offers one of the most comprehensive methods for evaluating feasibility and effectiveness.[30 31 40 41] For the feasibility assessment, women between 12 and 16 weeks gestational age and identified to be at risk for pre-eclampsia were recruited to self-monitor their BP using an automated BP monitor (Microlife WatchBP Home validated for use in pregnancy and pre-eclampsia). Study participants had the option to also text their readings and receive automated responses and text prompts.[31] Women reported that the approach was reassuring and made them more informed about HDPs; however, study authors noted the limited generalisability of the findings across different socioeconomic backgrounds.[31] This assessment was followed by a prospective cohort study of approximately 200 women and demonstrated that women at risk for pre-eclampsia were able to detect gestational hypertension in-between clinical visits.[30] The study also explored healthcare workers' perspectives and noted their concerns in reconciling the autonomy and reassurance that women may gain in exchange with the caution and uncertainty that self-monitoring requires.[41] More recently, the BuMP study conducted a randomised clinical trial (RCT) among 2441 pregnant individuals at increased risk for pre-eclampsia, in which there was no statistically significant differences in early detection of pre-eclampsia between SMBP and standard care groups.[42] Similarly, there was also no statistically significant differences among pregnant individuals with chronic or gestational hypertension.[43]

Another RCT, also conducted in the UK, enrolled pregnant women with chronic hypertension or gestational hypertension for SMBP using a calibrated BP measurement device (Microlife WatchBP Home); women with pre-eclampsia were excluded from this study.[44] Study participants reported their second of two BP readings via text, an application designed for the study or noted it on paper to be reviewed at their ANC contact.[44] In response to their text message or the mobile application, participants received immediate automated responses. Although there was no difference in the mean systolic BP measurements or medication adherence, participants demonstrated willingness and acceptability for self-monitoring.[44]

## Potential for digital technology to facilitate SMBP

Advances in digital technology in LMICs, primarily through penetration of mobile devices,[45] provide an opportunity to expand SMBP and other health monitoring services. While the catalyst for SBPM in high-income settings was spurred by the rise of automated cuffs,[12] this has not gained traction in settings in which there is already a limited availability of automated equipment for facility based use.[46] Furthermore, BP measurement devices require additional validation specific to pregnant populations.[37 47]

As an emerging alternative to measuring BP, OptiBP is a software application that uses smartphone cameras to capture optical pulse-waves of the fingertip and estimate BP readings (figure 1). The OptiBP application runs on Android OS V.8.1 and leverages the smartphone camera to record photoplethysmographic optical pulse waves derived from blood volume changes at the fingertips and estimate BP values using an algorithm.[48 49] This software

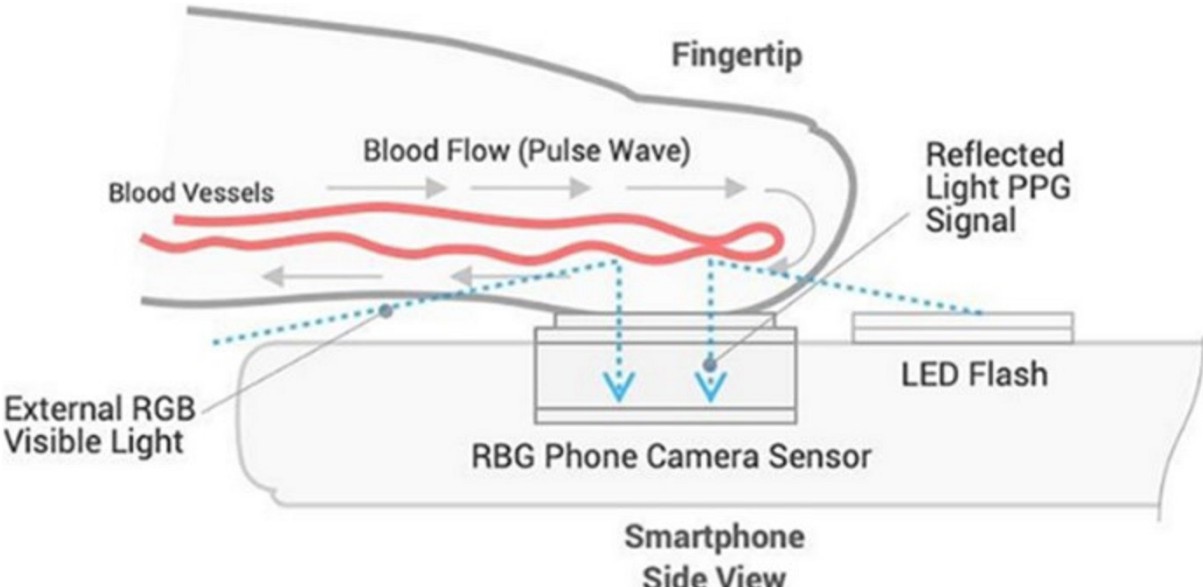

**Figure 1** Optical pulse wave technology underlying OptiBP smartphone application.[49] PPG, photoplethysmographic. LED, Light-Emitting Diode a form of light for flash on camera; RBG, red, blue, green for depicting color images on smartphones.

based algorithm has undergone validation in Switzerland, in which it demonstrated concordance against reference BP measurements[49 50]; a complementary validation study of OptiBP was conducted in accordance with the international regulatory standards in Tanzania, South Africa and Bangladesh. This study also included a targeted assessment on the accuracy of BP measurement among pregnant populations to account for the maternal cardiovascular changes on the detection of optical pulse waves.[51 52] Findings from this multi-site study demonstrated accuracy against International Organisation for Standardisation (ISO) benchmarks, including among pregnant populations, except in Bangladesh for systolic BP ISO criterion 2.[52 53] A similar validation study was also conducted in Indonesia among pregnant populations in West Lombok and East Lombok in West Nusa Tenggara Province.[54]

## Aims and objectives

The newfound capability of using a smartphone application for BP measurement presents an opportunity for broadening SMBP to LMIC settings, where the burden of HDPs is the greatest and mobile technologies have gained widespread use for maternal health.[51 55–63] Using these advances in BP monitoring, this research addresses evidence gaps identified in the WHO guideline on self-care interventions regarding the feasibility, benefits and harms of SMBP among individuals with HDP, particulary for non high income settings.[21] This study explores the perceptions, barriers and operational considerations for SMBP via a smartphone application (OptiBP) in Lombok, Indonesia. The first objective of the study is to examine pregnant women's, their partners and healthcare workers' attitudes towards SMBP in general and specifically regarding the use of a smartphone application for SMBP. In addition, we will analyze the implementation requirements for SMBP via a smartphone through

user-centred design approaches to adapt for health literacy and workflow integration requirements.

## METHODS AND ANALYSIS

### Study design and setting

This study will use a combination of qualitative methods, including focus group discussions (FGD), in-depth interview (IDI) and workshop observations to triangulate findings on perceptions and operational considerations for SMBP by pregnant women. The study will include pregnant women, their partners and the healthcare workers involved in the provision and management of ANC services to obtain a holistic understanding of the dynamics across health system actors. To derive concrete operational considerations, selected pregnant women will also be provided with a research version of the OptiBP smartphone application to use in their home setting and subsequently provide feedback on their experiences. The study will be conducted within the catchment area of six primary healthcare facilities in West Lombok district Lombok, Indonesia. Data collection began in mid-February 2023 and expected to be completed by May 2023.

### Participant selection

The main population for this study is pregnant women with current or past HDP. Eligible participants will be drawn from pregnant women coming to receive ANC at the selected study facilities and registered in the ANC digital module, which is a digital health record tracking and decision support system that was implemented independently of this study.[64] Following approval from the heads of health facilities to review the digital health records for sampling purposes, the study team will select pregnant women for screening and invitation to

participate in the study. A subset of the partners of participating pregnant women will be selected following the consent of the pregnant woman to contact her partner.

A diverse range of health workforce cadres, including community health workers, community and facility-based midwives, obstetrician/gynaecologists and health system managers will also be included to compliment the findings from pregnant women. Table 1 provides an overview of the participants and sampling frame.

## Sampling and recruitment

The study will use purposive sampling of pregnant women with a current diagnosis or history of HDP (as derived from their medical record in the ANC digital module) and across two age groups to account for potentially differing levels of digital literacy. We will identify women with different educational levels, gestational age and parity through the sampling and collect variables on these factors.

Pregnant women will be approached during their routine ANC contacts once they have completed their consultation. A research assistant will brief women on the study and gauge their interest to participate. The contact information of consenting and eligible pregnant women will be taken, and the research assistant will inform them that they will be contacted if they are selected for the study. Eligibility for pregnant women will be based on the following criteria: confirmed pregnancy (either by healthcare worker or self-administered and documented rapid test), current or history of HDP (derived from the digital record system), greater than 18 years of age or emancipated minor, living within catchment area of selected health facilities, able to read one sentence in Bahasa and having provided written informed consent. Pregnant women will be excluded if they have severe visual impairment that prevents use of the smartphone application (eg, blindness) or absence of index or middle finger on either left or right hand. Partners of pregnant women will be identified through the recruited women after first obtaining consent from the pregnant women to contact their partners. Pregnant women will not be excluded based on ownership of a phone, and the study will provide a smartphone to the participating women who are assigned to use the smartphone application at home.

Healthcare managers will be selected based on their role and oversight in the catchment area. They will be invited to participate in the study and asked for their written informed consent during their work hours. The different cadres of healthcare workers will be identified through District Health Office (DHO) and facility networks under its hierarchical systems, which are Public Health Centres (Puskesmas) and Maternity Clinics (Polindes).

Healthcare workers will be randomly selected to participate in the study based on a list provided by the DHO. The study team will first briefly inform the supervisors and invite the selected healthcare workers to attend an information session where the research staff will elaborate

**Table 1** Sampling frame

| Activity/method | Pregnant women with HDP | | Partners of pregnant women | Health workers/manager | | | | | |
| | <30 years | ≥30 years | | CHWs | Nurse-midwives (community) | Nurse-midwives (facility) | Doctors | Health system managers |
| --- | --- | --- | --- | --- | --- | --- | --- | --- |
| FGD on perception of SMBP | 2–3 FGDs of 5–7 | 2–3 FGDs of 5–7 | 2 FGDs of 5–7 | 4 FGDs of 5–7 | 2–3 FGDs of 5–7 | 2–3 FGDs of 5–7 | N/A | N/A |
| IDI on perception of SMBP | 4–5 | 4–5 | N/A | 4–5 | 4–5 | 4–5 | 3–5 | 3–5 |
| Intervention adaptation workshops | 5–10 | 5–10 | N/A | N/A | N/A | N/A | N/A | N/A |

CHWs, community health workers; FGD, focus group discussion; HDP, hypertensive disorders of pregnancy; IDI, in-depth interview; N/A, not applicable; SMBP, self-monitoring of blood pressure.

further on the study and ask for their consent to take part in the study. The interviews will take place during work hours after permission has been obtained from the DHO.

The study will invite women to either be part of the FGDs or workshops and randomly allocate to the two groups. Participants for the IDIs will be sampled from the FGDs to examine their feedback in greater detail. The sample size for pregnant women and healthcare workers will be based on requirements for reaching saturation, as well as using an emergent sampling design, in which participants will be recruited as the study progresses.[65]

## Methods of approach
### Interviews
FGDs and IDIs will be used to examine potential barriers and motivators for SMBP, as well as how this may influence changes in pregnant women's interactions with healthcare workers. As partners and family members of pregnant women can influence access and use of ANC services, their perceptions of SMBP will also be used to obtain a comprehensive understanding of the considerations for SMBP and any related care.[66]

### Observation workshops
In addition to the FGDs and IDIs, observation workshops will be conducted to review the smartphone application interface and determine adaptation requirements for enhancing its usability in accordance with health literacy levels and ensuring integration with health system and data workflows. The workshop will have two parts. First, we will introduce a research version of the smartphone application to selected women and observe initial reactions and feedback. Women will be provided with a smartphone loaded with the application and instructions to explore using it at home twice a day for 1 week before returning to the subsequent workshop, 1 week later. This second workshop will be used to gather feedback on their experiences of use and refine guidance/training support needed for achieving fidelity and requirements for integrating within health system workflows.

The software application provided to the participants will be linked to a de-identified study account with access to the local research team. It will also include a disclaimer stating the measurements are not to be used for clinical decision-making and that participants should contact the health facility and research team if they face any health issues. The application will not be used by the research team or healthcare workers for any clinical decision-making; all BP measurements for service provision will be done through the standard care at facilities.

Both workshops will include structured and unstructured observations of pregnant women engaging with the smartphone application interface and will be used to further inform the decision support prompts (eg, what to do in cases of high BP reading) that are contextually appropriate and resonate with the levels of health literacy. In the structured observations, pregnant women will be asked to interact with the smartphone application prototype across various tasks and scenarios, such as turning on the smartphone application, placing their finger on the camera of the smartphone for taking their BP measurement, interpreting the result/output from the BP measurement and undertaking follow-up actions, when necessary.

## Data collection
For the interviews, a semi-structured topic guide will be provided to trained interviewers conducting the FGDs and IDIs. The topic guide for pregnant individuals will cover areas of pregnancy experience, mobile phone use in pregnancy, knowledge of and experience with taking BP, self-efficacy in taking BP, trust and support for taking BP. Interview guides for the healthcare worker will include topics surrounding experience in providing ANC, perceptions of self-management of pregnancy generally, perceptions of SMBP by pregnant women and data flow and health system linkages. All interview guides will include probes and questions for initiating the interviews and allow for open-ended responses and capturing any reflections that may emerge. Interviews will be recorded, transcribed and translated into English for analysis.

For the observation workshops, a research assistant will document both unprompted reactions, as well as responses to specific tasks with the smartphone application. As a supplement to the observation notes, the study will also embed a screen recorder while pregnant women use the application during the workshop to capture the process flows and interactions with the application, including length of time and frequency of use of different buttons. The study team will also take videos and photos during the observation workshop to record feedback related to interactions with the application.

## Analysis
Recordings will be double transcribed and translated for quality assurance. Data obtained from the interviews and observations will first undergo an initial rapid analysis to flag key themes and considerations, including findings from the existing literature base. This will be followed by detailed analysis and coding using a thematic analysis employing a combination of both inductive and deductive approaches.[65 67] Each finding will be compared and cross-checked to determine whether it aligns with existing categories or if a new theme should be developed.[65] The analysis will be conducted by two individuals to ensure inter-rater reliability and then reviewed collectively with the broader research team. Screen recordings of interactions with the application will also be reviewed to identify patterns of use. ATLAS.ti[68] will be used to manage the coding and analysis of themes from the transcripts.

## Patient and public involvement
The views of patients or the public will be reflected in the analysis of the manuscript and through the co-design workshops of the study.

## Ethics and dissemination

The study was approved by the WHO/Human Reproduction Programme Research Review Panel and WHO Ethical Review Committee (Reference number A65932) as well as the Health Research Ethics Committee, Faculty of Medicine, Universitas Mataram in Indonesia (004/UN18/F7/ETIK/2023).

All eligible participants will receive an information sheet and are required to provide informed written consent before participation. Confidential data will be stored at country level and de-identified transcriptions will be shared with WHO to support the analyses. Personal identifiers will be removed during the transcription of audio recordings and labelled with a de-identified participant identification number. Identifiable data will be kept at the research site in a secure location for a minimum of 5 years before securely disposing of primary research data, as per the guidance from the local ethical review committee.

Participants using the smartphone application at home will be provided with detailed guidance, as well as airtime/SIM cards to contact the research team as needed. The smartphone application will not be used by the research team or healthcare workers for any clinical decision-making, and all BP measurements for service provision will be done through the standard care at facilities using standard sphygmomanometers.

Photos and video recordings will only be taken during training and specific group events, and only after written informed consent. The informed consent will explicitly mention the use of audio-recording and screen recording during the workshop. Unless otherwise consented, use of these will be obscured by blurring or other effects to render persons non-identifiable, and the purpose for use will be restricted to training and implementing processes to make the work or intervention more effective. The company that developed the smartphone application will not have access to any data or transcripts but will have access to the feedback shared during their participation in the workshop.

Findings will be disseminated through research publications and communicated to the DHO. The analyses from this study will also inform the design of a subsequent impact evaluation. In addition, the feedback from workshops may contribute to potential refinements to the smartphone application.

**Author affiliations**
[1]UNDP/UNFPA/UNICEF/WHO/World Bank Special Programme of Research Development and Research Training in Human Reproduction (HRP), Department of Sexual and Reproductive Health and Research, World Health Organization, Geneva, Switzerland
[2]University of Geneva, Geneva, Switzerland
[3]Summit Institute for Development, Mataram, Indonesia
[4]Brawijaya University, Malang, Indonesia
[5]HI5lab, Faculty of Medicine, University of Geneva, Geneva, Switzerland
[6]Oxford University Clinical Research Unit-Indonesia, Jakarta, Indonesia

**Acknowledgements** We thank the heads of health centres from the study sites (Penimbung, Ranjok, Jembatan Kembar, Narmada Gerung and Meninting) for their support to conduct the study. We are also grateful to Dr Garrett Mehl (WHO) for support in conceptualisation of the study.

**Contributors** TT, YDS, MB, AHS and ÖT developed the protocol from which excerpts were used to draft this manuscript. AG, MG, MPP and SOR reviewed the protocol and provided comments that fed into the manuscript. TT prepared the first draft of the manuscript. All authors reviewed both the protocol and this manuscript. ÖT and AHS are shared senior authors.

**Funding** This work was funded by the Bill and Melinda Gates Foundation (grant number OPP1201339) and the UNDP-UNFPA-UNICEF-WHO-World Bank Special Programme of Research, Development and Research Training in Human Reproduction (HRP), a cosponsored programme executed by the WHO. This research received no specific grant from any funding agency in the public, commercial or not-for-profit sectors.

**Disclaimer** The authors alone are responsible for the views expressed in this article and they do not necessarily represent the views, decisions or policies of the institutions with which they are affiliated.

**Competing interests** None declared.

**Patient and public involvement** Patients and/or the public were not involved in the design, or conduct, or reporting, or dissemination plans of this research.

**Patient consent for publication** Consent obtained directly from patient(s).

**Ethics approval** The study was approved by the WHO/HRPHuman Reproduction Programme Research Review Panel and WHO Ethical Review Committee (Reference number A65932) as well as the Health Research Ethics Committee, Faculty of Medicine, Universitas Mataram in Indonesia (004/UN18/F7/ETIK/2023).

**Provenance and peer review** Not commissioned; externally peer reviewed.

**ORCID iD**
Tigest Tamrat http://orcid.org/0000-0001-8579-5698

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
