## [Reviewer comments · BMJ Open]

ARTICLE DETAILS

TITLE (PROVISIONAL)	Exploring perceptions, barriers and facilitators for self-monitoring of blood pressure in pregnancy using a smartphone application in Lombok, Indonesia: Protocol for a qualitative study
AUTHORS	Tamrat, Tigest; Setyawati, Yuni Dwi; Barreix, Maria; Gayatri, Mergy; Rinjani, Shannia Oktaviana; Pasaribu, Melissa Paulina; Geissbuhler, Antoine; Shankar, Anuraj H; Tunçalp, Özge

VERSION 1 – REVIEW

REVIEWER	Ellis, Shellie University of North Carolina at Chapel Hill Gillings School of Global Public Health, Health Policy and Management
REVIEW RETURNED	20-Apr-2023

GENERAL COMMENTS	The protocol is well described. There are some areas of redundancy (e.g., target populations) that could be edited out for conciseness.
---

REVIEWER	Helou, Amyna Monash University, Centre for Medicine Use and Safety
REVIEW RETURNED	25-Sep-2023

GENERAL COMMENTS	Overall well written qualitative study protocol, commendable for publication as a study protocol after clarification of study dates. The illustration of need for such a project is demonstrated well especially in comparison to high income vs. low and middle income countries. Some specific comments for author consideration include:  - On page 4, lines 12-14 in the Introduction, there is reference to pre-eclampsia occurring in pregnant women with gestational hypertension but no reference to pre-eclampsia also occurring in pregnant women with chronic hypertension. Please consider including this as pre-eclampsia superimposed on chronic hypertension is a cause of potentially severe morbidity and mortality. - On Page 5, line 40 of Introduction, please include a reference for line ending: “validation studies towards regulatory approval are in progress.” - In regards to Methods; if the study participants do not have access to an appropriate Android phone, will this be a point of study exclusion or will the study provide a phone for the study?
--

	 - Methods; page 7 line 35, insert 'written' to informed consent and mention that the informed consent will include consent for audio-recording and screen recording for the workshops - In Analysis; page 9, please provide a reference for Atlas.ti - Ethics and dissemination: page 9 line 36, what happens to the data after 2 years? Please detail - References; please complete the full citations for the following references: - 16 Niiranen TJ, Jula Am Fau - Kantola IM, Kantola Im Fau - Reunanen A, Reunanen A. Comparison of agreement between clinic and home-measured blood pressure in the Finnish population: the Finn HOME Study - 26 Bray EP, Holder R, Mant J, McManus RJ. Does self-monitoring reduce blood pressure? Meta-analysis with meta-regression of randomized controlled trials. Database of Abstracts of Reviews of Effects (DARE) - 49 Ghamri Y, Proença M, Hofmann G, Renevey P, Bonnier G, Braun F, et al. Automated Pulse Oximeter Waveform Analysis to Track Changes in Blood Pressure During Anesthesia Induction: A Proof of-Concept Study - 52 Tigest Tamrat. Charles Festo VV, Tsakane Hlongwane, Hasmot Ali, Getrud Mollel, Kaniz Fahmida, Kelsey Alland, Maria Barreix, Hedieh Mehrtash, Ronaldo Silva, Soe Soe Thwin, Garrett Mehl, Alain Labrique, Honorati Masanja, and Özge Tunçalp,. Accuracy of a smartphone application for blood pressure estimation in Bangladesh, South Africa, and Tanzania. NPJ Digital Medicine Also reduce author list to 5 then et al. to have consistency in referencing  - 63 Kostanjsek Nea. Use of ICD-11 and other WHO classifications and terminologies in clinical and public health guideline development and implementation
--	--

VERSION 1 – AUTHOR RESPONSE

Dear Editor/Reviewers,

Thank you for your valuable feedback and constructive feedback to strengthen this manuscript. Please see below our responses in blue to the comments received.

Some specific comments for author consideration include:

- On page 4, lines 12-14 in the Introduction, there is reference to pre-eclampsia occurring in pregnant women with gestational hypertension but no reference to pre-eclampsia also occurring in pregnant women with chronic hypertension. Please consider including this as pre-eclampsia superimposed on chronic hypertension is a cause of potentially severe morbidity and mortality.

Thank you for noting this. We have reviewed this section and included the following phrase to be inclusive of chronic hypertension with superimposed preeclampsia:

“Hypertensive disorders of pregnancy include chronic hypertension, gestational hypertension, preeclampsia, and chronic hypertension with superimposed preeclampsia.”

- On Page 5, line 40 of Introduction, please include a reference for line ending: “validation studies towards regulatory approval are in progress.”

Thank you. We have included the following references:

- Festo C, Vannevel V, Ali H, Tamrat T, Mollel GJ, Hlongwane T, et al. Accuracy of a smartphone application for blood pressure estimation in Bangladesh, South Africa, and Tanzania. *npj Digital Medicine*. 2023;6(1):69.
 - Setiyawati YD, Jannah M, Gayatri M, Meylentina E, Shankar AH (In Press) The accuracy of cuffless optical blood pressure assessment in pregnancy and its potential to improve clinical outcomes.
-
- In regards to Methods; if the study participants do not have access to an appropriate Android phone, will this be a point of study exclusion or will the study provide a phone for the study?
Thank you for raising this point. Phone ownership will not be a point of exclusion as the study will provide a phone for the pregnant women who are assigned to the group that will be using the application in their home setting. We have added this clarification in the methods.
 - Methods; page 7 line 35, insert ‘written’ to informed consent and mention that the informed consent will include consent for audio-recording and screen recording for the workshops
Thank you for this feedback. We have added this to the methods.
 - In Analysis; page 9, please provide a reference for Atlas.ti
We have added this reference.
 - Ethics and dissemination: page 9 line 36, what happens to the data after 2 years? Please detail
Thank you for raising this point. The WHO IRB requires data to be stored for a minimum 2 years. However, the materials will be retained for a minimum of 5 years as per local IRB requirements before securely disposing of the original research data. We have added this clarification in the manuscript.
 - References; please complete the full citations for the following references:
Thank you for noting this. We have updated the citations accordingly.
 - 16 Niiranen TJ, Jula Am Fau - Kantola IM, Kantola Im Fau - Reunanen A, Reunanen A. Comparison of agreement between clinic and home-measured blood pressure in the Finnish population: the Finn HOME Study
 - 26 Bray EP, Holder R, Mant J, McManus RJ. Does self-monitoring reduce blood

pressure? Meta-analysis with meta-regression of randomized controlled trials.
Database of Abstracts of Reviews of Effects (DARE)

- 49 Ghamri Y, Proença M, Hofmann G, Renevey P, Bonnier G, Braun F, et al. Automated Pulse Oximeter Waveform Analysis to Track Changes in Blood Pressure During Anesthesia Induction: A Proof of-Concept Study
- 52 Tigest Tamrat. Charles Festo VV, Tsakane Hlongwane, Hasmot Ali, Getrud Mollel, Kaniz Fahmida, Kelsey Alland, Maria Barreix, Hedieh Mehrtash, Ronaldo Silva, Soe Soe Thwin, Garrett Mehl, Alain Labrique, Honorati Masanja, and Özge Tunçalp,. Accuracy of a smartphone application for blood pressure estimation in Bangladesh, South Africa, and Tanzania. NPJ Digital Medicine

Also reduce author list to 5 then et al. to have consistency in referencing

We have removed the following citation.

- 63 Kostanjsek Nea. Use of ICD-11 and other WHO classifications and terminologies in clinical and public health guideline development and implementation